

# An improved statistical bias correction method that also corrects dry climate models

Fabian Lehner[1], Imran Nadeem[1], and Herbert Formayer[1]

[1]Institute of Meteorology and Climatology, Department of Water, Atmosphere and Environment, University of Natural Resources and Life Sciences (BOKU), Gregor Mendel Straße 33, 1180 Vienna, Austria

**Correspondence:** Fabian Lehner (fabian.lehner@boku.ac.at)

**Abstract.** Daily meteorological data from climate models is needed for many climate impact studies, e.g. in hydrology or agriculture but direct model output can contain large systematic errors. Thus, statistical bias correcting is applied to correct the raw model data. However, up to now no method has been introduced that fulfills the following demands simultaneously: (1) The long term climatological trends (climate change signal) should not be altered during bias correction, (2) the model data should match the observational data in the historical period as accurate as possible in a climatological sense and (3) models with too little wet days (precipitation above 0.1 mm) should be corrected accurately, which means that the wet day frequency is conserved. We improve the already existing quantile mapping approach so that it satisfies all three conditions. Our new method is called empirical percentile-percentile mapping (EPPM) which uses empirical distributions for meteorological variables and is therefore computationally inexpensive. The correction of precipitation is particularly challenging so our main focus is on precipitation where EPPM corrects the historical model data so that precipitation sums and wet days are equal to the observational data.

## 1 Introduction

Daily data from climate models are used for various applications, e.g. in hydrology, silviculture and for general climate risk studies (e.g. Horton et al., 2017; Seidl et al., 2019). However, simulated outputs from global climate models (GCMs) and regional climate models (RCMs) can exhibit large systematic biases relative to observational data sets (Mearns et al., 2013; Sillmann et al., 2013). These systematic errors can be statistically corrected with gridded observations. Those corrected data sets are widely used (e.g. Bao and Wen, 2017; Thrasher et al., 2012; Chimani et al., 2016) but are controversial due to various errors introduced by statistical correction. In the last two decades a series of methods for statistical bias correction were presented:

A very simple approach is to correct the mean of the observations by adding the climate change signal "CCS" found in the model (i.e. the change of a meteorological variable over time) to the observations (Maraun, 2016), usually referred to as delta change correction. Due to its simplicity, it is still in use (Navarro-Racines et al., 2020). Another simple yet conceptually different approach is the mean bias correction where the present-day model bias is subtracted from the future time series. This linear off-set method was used in Lafon et al. (2013). Higher moments of a distribution like variance and skewness in the



observational data remain unchanged though these higher moments may be of importance in climate risk studies, where the exceedances of specific thresholds of temperature or precipitation are often relevant. Alternatively, as a different concept, one can find the bias of the model by calculating the difference of the means between the observations and the model. This bias is then applied to the future time periods by only changing the mean.

A more sophisticated approach is to change every percentile of the cumulative distribution functions (CDF) according to
the differences between daily modeled and observational data during a reference period. There are many different variations and names for this method in the literature: variable correction method (Déqué, 2007), distribution-based scaling (Yang et al., 2010; Seaby et al., 2013), distribution mapping (Teutschbein and Seibert, 2012), statistical bias correction (Piani et al., 2010), statistical transformation (Gudmundsson et al., 2012), quantile-quantile mapping (Hatchett et al., 2016; Potter et al., 2020; Charles et al., 2020) or quantile mapping (QM) (Li et al., 2010; Themeßl et al., 2011; Maraun, 2016). QM usually outperforms
other simpler methods like mean bias correction as shown by Lafon et al. (2013) or Themeßl et al. (2011). The distribution of the meteorological variables can be described with empirical CDFs which is a non-parametric approach. Many QM methods use a parametric approach instead, where statistical functions like gamma or normal distributions are fitted to the CDFs.

However, traditional QM may alter the raw CCS of the climate model (Maurer and Pierce, 2014; Maraun, 2013, 2016) and therefore even introduce new biases that are comparable to the biases of the original model. To better preserve the CCS of the
models, modified QM methods were developed over the past years. One of them is detrended quantile mapping (DQM) (Bürger et al., 2013), where the change in mean is removed before bias correction with traditional QM. Similar to DQM, Hempel et al. (2013) removed the change in mean before applying a form of parametric QM. However, the QM in Hempel et al. (2013) is done with a linear regression (i.e. the model bias decreases or increases linearly with the variable e.g. temperature or precipitation, so there is no exact correction for each quantile)

Moreover, DQM method does not preserve the climatological trend of extremes. Extending previous work, Cannon et al. (2015) developed an approach termed as quantile delta mapping (QDM). A very similar approach termed the equidistant CDF matching method (EDCDFm) was introduced by Li et al. (2010). QDM respectively EDCDFm compare the CDF of the historical and future model data. For each percentile, this difference is saved either as absolute difference (for temperature and dew point) or as a ratio (for most variables like precipitation, wind speed, radiation). These values describe the CCS for each
percentile of the CDF. Finally, these values are applied on the observations to get to the bias corrected future model.

Recently, Charles et al. (2020) applied QM on precipitation and extended QM with an additional rescaling after bias correction to match the raw model's relative CCS . With this forced CCS, any (monthly, seasonal or annual) CCS of the raw model can be reproduced with high accuracy.

Another method is the multi-segment statistical bias correction (MSBC) (Grillakis et al., 2013) optionally combined with
a normalization module (NM) (Grillakis et al., 2017). With this QM technique, in a first step every year's CDF is transferred to the reference period's CDF to remove long term trends. In a second step, the CDFs of both model and observations are approximated by piecewise functions (making MSBC a parametric method) which has the ability to better represent the original CDF than one single fit to the entire CDF. However, both studies did not compare with other bias correcting methods, so the actual benefit compared to traditional QM remains unknown.





Switanek et al. (2017) developed a new method called scaled distribution mapping (SDM), which fits parametric distribution functions to the data. SDM is based on QDM. The authors found a better representation of extreme values with SDM. One of the key focuses of this work was the accurate correction of the number of wet days, which are defined as days with a precipitation above a threshold. In this work, the threshold is equal to 0.1 mm. A downside of this method is the relatively high demand of computational power, as the parameter fitting is iteratively done for each time step and each grid point. Furthermore, this

method does not take cases into account where the model has less wet days than the observation, which can lead to substantial biases in precipitation sums in some very dry climate models.

In general, all statistical bias correcting methods that use CDFs, can be grouped by following characteristics:

– Direct or indirect approaches (Maraun, 2016): The first one directly corrects the model output via the model bias, also referred to as "bias correction" (Ho et al., 2012). For all direct methods we assume a temporal stationarity of the model

bias to be able to apply the bias to future model data, even though this assumption may not be fully valid (Chen et al., 2013; Ho et al., 2012; Teutschbein and Seibert, 2012). The indirect methods are applied to the observational data via adding the model's CCS (in Ho et al. (2012) named as "change factor").

– Quantile of the corrected result: The methods differ concerning to the quantiles of the raw model and the corrected model. Most methods stay on the same quantiles after correction. In only a few methods the result is on a different

quantile with one of those being CDF-t (Michelangeli et al., 2009; Pierce et al., 2015).

– Location of the correction term: The correction value can either be taken from the current quantile or from the corresponding absolute value of a variable (e.g. 0.6 quantile or alternatively 10 mm absolute value). Methods that use absolute values may need extrapolation at both ends of the CDF for values that were not observed in the observational time period.

– Last, some methods use a parametric approach, i.e. they fit functions to the distribution of variables (e.g. Hempel et al.,

2013; Piani et al., 2010; Switanek et al., 2017), while other methods use empirical distribution functions that is sampled with discrete intervals (e.g. Cannon et al., 2015).

Whether to use a nonparametric or a parametric approach is still in scientific discussion (Teng et al., 2015) but the non-parametric approach is more common. Lafon et al. (2013) compared nonparametric (empirical) and parametric QM where the empirical one is the most accurate. Also, Cannon et al. (2015) and Gudmundsson et al. (2012) prefer the empirical QM. The-

meßl et al. (2012) point out that parametric QM can introduce new biases, because the distribution of a meteorological variable is not fully known and also depends on the region and season. However, nonparametric QM depends more on the calibration period than parametric QM. Switanek et al. (2017) argue that the correction of extremes is more robust if one uses a parametric approach, as the return level of the most extreme event is somewhat random. This can be improved by fitting function to the distributions.

Most of the bias correcting methods correct the wet day bias of climate model (i.e. the number of wet days above a specific precipitation threshold) only if the model has a positive wet day bias. The overestimation of wet days is very typical for dynamical models and is known as the "drizzling effect" (e.g. Gutowski Jr. et al., 2003). However, in some rare cases the model





may have too little wet days which have to be corrected separately. Only very few studies have focused on correcting a negative wet day bias, with one of them being Themeßl et al. (2012). They use a simple linear interpolation to fill the gap of wet days in

the precipitation CDF. This removes the wet day bias but does not necessarily conserve precipitation sums, because the CDF of precipitation does not follow a linear curve.

However, up to now no method has been introduced that fulfills the following demands simultaneously: (1) The long term climatological trends should not be altered during bias correction, (2) the model data should match the observational data in the historical period and (3) models with too little wet days should be corrected accurately which means that dry days have to

be set to wet days. (2) and (3) are often neglected in comparison to (1).

Almost all bias corrections are capable of (2) for temperature, but for precipitation new challenges arise: First, parametric methods struggle to find an accurate function for daily precipitation values and second, (2) and (3) are linked, i.e. in models with loo little wet days there are not enough wet days to correct and thus the precipitation sums are too low. Inaccurately corrected model data can cause wrong conclusions from climate impact studies, when the future climate data is compared to

the historical data. We found that still after bias correction the data is biased (e.g. Chimani et al., 2016; ClimaProof; CCCA data server).

We present an improved method based on QM called empirical percentile-percentile mapping (EPPM). The basic assumption of EPPM is that the model bias at each quantile or percentile is constant over time, either defined as absolute or relative bias. Therefore, in contrast to traditional QM, EPPM corrects merely on constant percentiles, i.e. the correction value and the bias

corrected model value are at the same percentile as the raw model's value. Thus, there is no need for an extrapolation method for new extremes in the future model. The assumption of climate stationarity like in traditional QM (Maurer and Pierce, 2014; Switanek et al., 2017) is not necessary for EPPM.

EPPM uses a nonparametric approach, and can therefore be applied to every meteorological variable without knowing the statistical distribution. This makes EPPM computationally inexpensive and usable for large domains or very high spatial

resolutions.

## 2   Data and area of interest

This study focuses on Austria which is located in Central Europe. The topography is shown in Fig. 1. A large part of the Eastern Alps are within the Austrian borders. The elevation reaches from 114 m in the East of Austria to 3798 m amsl on the highest mountain. Because of the complexity of the topography, the spatial resolution of GCMs and also RCMs is not sufficient to

resolve mountain ridges and valleys. The climatologic properties can change within a few kilometers due to topographically induced effects (Stauffer et al., 2017).

Austria has a large number of high quality weather observations that are operated by Zentralanstalt für Meteorologie und Geodynamik (ZAMG). Also, gridded observational data sets called SPARTACUS for minimum temperature, maximum temperature and precipitation are available on a daily basis on a high spatial resolution of 1 km (Hiebl and Frei, 2016, 2018). The





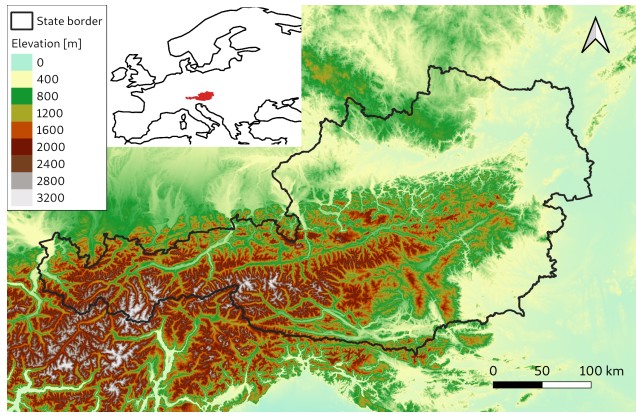

**Figure 1.** Area of interest with Austrian state borders. © European Union, Copernicus Land Monitoring Service 2020, European Environment Agency (EEA).

time span reaches from the year 1961 to 2019. SPARTACUS mostly uses stations with long time series to provide robust trends for climate change.

For the observational data, SPARTACUS in its unchanged form is used (thereafter named OBS). For model data, synthetic data is produced by smoothing SPARTACUS data with a running mean of 12 km. This is a typical spatial resolution of RCMs.

To get artificial dry model data, the data was further manipulated. This was done by multiplying the precipitation of each day

with a uniformly distributed random number between 0 and 1. Furthermore, a trend to even drier conditions was introduced by successively canceling more and more wet days going from 1961 to 2019. This artificial model data represents a dry model with too little wet days and too low precipitation sums.

To show that the bias corrected historical model data do not always match the observations, we analyzed data sets in Austria from the projects ÖKS15 (Chimani et al., 2016) and STARC-Impact (Chimani et al., 2019). In both projects, the model data

was bias corrected with SDM (Switanek et al., 2017). The data is freely available via the Climate Change Center Austria (CCCA) and consists of bias corrected temperature and precipitation data from several RCMs at a spatial resolution of 1 km. The data is used for many climate impact studies in Austria (e.g. Jandl et al., 2018; Unterberger et al., 2018).

We calculated climatological annual precipitation sums for all models in ÖKS15 in the reference period 1971-2000 and compared it with OBS. Fig. 2a shows the bias of the domain average annual precipitation for each model. The mean bias

ranges from approx. -2 % for the driest model to almost +6 % for the wettest model. The comparison on a grid cell basis on the right side in (Fig. 2b) shows biases of more than +50 % for the wettest 0.1 %, and a bias of approx. -25 % for the driest grid cells. Outliers can reach up to +100 % compared to the observational precipitation (not shown). The median is approx. +4 %.




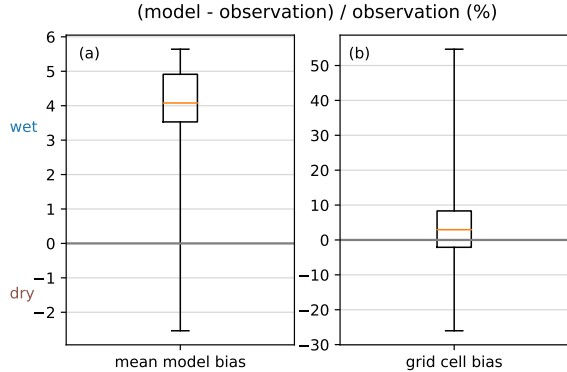

**Figure 2.** Box and whisker plot for the relative annual precipitation bias (%) of the ÖKS15 and STARC-Impact models (a total of 35 models) for the reference period 1971-2000. (a): Relative bias for the area mean for each climate models in Austria. (b): Relative bias on a grid cell basis. The upper (lower) whisker shows the 99.9 (0.1) percentile. The box ranges from the 25 to the 75 percentile, the orange horizontal line is the median.

## 3 Methods

Our study focuses on implementing a new method to bias correct data from climate models called EPPM. The method corrects

the CDF of a meteorological variable from a climate model to match the CDF of the observation. Each grid cell of the model is separately corrected with the observations on a monthly or seasonal basis. For the calibration data, a time interval of 30 years is typical, since the statistical distribution of data of a shorter time period can be very noisy and a longer time period usually has climatological trends.

### 3.1 Empirical percentile-percentile mapping (EPPM)

The basic assumption is that the model bias stays constant over time for each quantile or percentile of the model data. The terms quantile and percentile are almost synonymous, the only difference is that quantile uses a range of 0 to 1, whereas percentiles range from 0 to 100 (percent). In accordance with the name of the method we use the term percentile.

In other words, we consider that the RCM is able to predict a ranked category of temperature or precipitation but not the value for this variable (Déqué, 2007).

EPPM is divided into two parts. In the first part, the model bias is evaluated for a distinct number of quantiles of a variable's CDF, thereafter named correction values (CVs). In the second part, the correction values are applied to any desired time interval.

Step (1): If the variable is not limited to non-negative values (like temperature or dew point), detrending should be applied to the data before any further calculation is done. The 30 year data should be detrended for each month separately by subtracting a linear trend. This trend is added again after bias correction. Without detrending, the CCS can be altered within the bias

correction.





Step (2): If the variable of interest is precipitation, the further procedure depends on the difference in wet days between the model and the observation. If the model data has more wet days, all data is used. If the model data has less wet days than the observational data (which is the rare case), then only wet days are used for further calculation. The threshold for wet days are typically set at 0.1 mm precipitation per day.

Step (3): For the present-day calibration period, the empirical CDF (ECDF) is calculated for the model and the observational data. As data from 30 years for one single month is typically considered, there is a total of approximately 900 days for the estimation of the CDF. The number of discrete values for the ECDF is usually set to 100 (equidistant from 0.5 % to 99.5 %) which results in 100 CVs The number of 100 points was found via trial and error. A higher number of correction values would be less robust, as especially the CVs of extremes would depend even more on single extreme events.

Step (4): The correction values (CVs) are calculated. The CVs for parameters like temperature and dew point are calculated as difference between the observational $\text{CDF}_{\text{obs}}$ and the model $\text{CDF}_{\text{modp}}$ for the present-day climate:

$$CV = \text{CDF}_{\text{obs}} - \text{CDF}_{\text{modp}} \tag{1}$$

For parameters that have a meteorologically meaningful zero value that is the lower boundary, a multiplicative approach is more useful, e.g. for precipitation, wind speed or global radiation. The CVs for those parameters are found by

$$CV = \text{CDF}_{\text{obs}}/\text{CDF}_{\text{modp}} \tag{2}$$

If the model wind speed and global radiation should ever reach exact zero, Eq. 2 would not be defined. For this case. the CV is manually set to 0. For precipitation, one has to distinguish between models with too many or too little wet days (described in step (2) above and further information in Sect. 3.2 below). The CVs can be interpreted as the model bias for any quantile of the model data at a given grid cell.

Step (5): For extreme data at both ends of the distribution, the CVs have to be extrapolated. This is done via constant extrapolation, i.e. the first (last) CV is used for correcting data below (above) the outermost CDF value. All model data values below the 0.5 % percentile are corrected with the CV attached to the 0.5 % percentile and model data values above the 99.5 % percentile are corrected with the CV attached to the 99.5 % percentile.

     Step (6): Any desired time period is selected where the CDF is calculated for the model data. It is possible to choose the

calibration period itself. Again, if the variable is not limited to non-negative values, the linear trend has to be removed from the model data to avoid altering the CCS and added again after bias correction. The time period to be chosen is usually a 30-year period like in the calibration time period.

     Step (7): The CVs are added (for temperature and dew point) or multiplied (for precipitation, global radiation and wind speed) to the model CDF calculated in step (5). This gives the CDF of the bias corrected model. The correction of the original

time series of the raw model is done by interpolating the time series from the original CDF from step (5) to the corrected CDF from this step. As the CDF is only defined at discrete values in an ECDF, data in between is linearly interpolated.

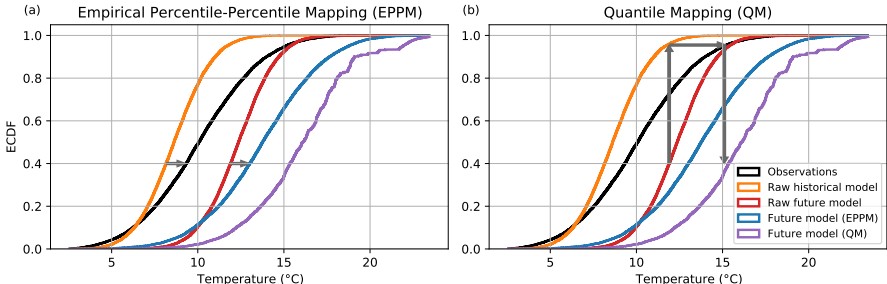

**Figure 3.** Schematic of bias correction for temperature data. CDFs are shown for following data: Observational data (black), raw historical model (orange), raw future model (red), future model corrected with EPPM (blue) and future model corrected with QM (purple). The arrows illustrate the bias correction path for future model data. Panel (a) shows EPPM, where the model bias in the calibration period (left arrow) is applied to the future model data (right arrow that has the same length as the left arrow). Panel (b) shows QM, where the correction value is found at the model bias in the historical period of the same absolute value.

Step (8): If the data was detrended in step (6), the trend has to be added to the model data.

Step (9): Even after detrending, there could still be a non-linear trend in the data. Therefore, not all 30 years that were used during bias correction are saved, but only the middle 10-year period. For this reason, the bias correction has to be calculated in 10-year steps.

The graphical solution for bias correction for temperature data is shown in Fig. 3. The temperature data in this plot is artificially created following a normal distribution, where the historical period is 1981-2010 and the future period is 2071-2100. In this hypothetical case the present-day model has a significant bias in mean and variance:

- The observational data features a mean of 10 °C and a standard deviation of 3.1 °C.

- The raw historical model (1981-2010) has a cold bias in the data with a mean of 8 °C and a standard deviation of 1.8 °C.

- The raw future model (2071-2100) is warmer with a mean of 12.4 °C but the standard deviation remains unchanged to the historical model with a standard deviation of 1.8 °C.

For the raw historical model, these distributions result in too low temperatures in the upper end of the CDF and slightly too high temperature in the lower end. During the bias correction of EPPM, this model bias of each quantile is added to the future model resulting in the bias corrected model (purple line). In contrast, QM uses equivalent model values from the historical period. As during climate change higher temperatures occur more often, correction values from the upper part of the CDF are used more often. As higher temperatures tend to have larger biases in the raw historical model, the correction with QM results in a too warm bias corrected model.



### 3.2 Precipitation: adding wet days after EPPM (EPPMd)

EPPM (along with other methods like QM and SDM) corrects by default the number of wet days if the model has more wet days than the observational data by multiplying the lower parts of the model CDF with 0. However, EPPM can not add wet days that are initially not in the model. Therefore, an extension of EPPM called EPPMd (d for dry mode) is developed that adjusts the wet day frequency of the model to match the observational wet day frequency. As second condition, the amount that is added should reproduce the precipitation sum of the observational data.

The model bias in wet day frequency is defined as an absolute one, i.e. the number of wet days that have to be added to the model data. This is more practical regarding extremely dry models or extremely dry climates. For example, if there are only 3 wet days per month in the calibration period of model data but 12 wet days in the observational data, the absolute difference is 9 days. In contrast, the relative bias is 400 %. In a future time period the model data may have 7 wet days per month. The absolute bias correction adds 9 additional wet days to a total of 16 wet days per month, while the relative bias correction would

add 400 % and lead to 28 wet days per month. Eventually, an absolute definition of the model wet day bias leads to more robust results.

Most climate models have the Gregorian calendar with 365 or 366 days a year. In contrast, a few models have a 360-day calendar, where each month is exactly 30 days. As the observations are always according to the Gregorian calendar, a simple absolute difference in wet day frequency is not always reasonable. Thus, a normalized wet day frequency is defined as the ratio

between wet days and all days. The values ranges from 0 (only dry days) to 1 (only wet days).

If the model has less wet days than the observations, EPPM only corrects the wet model days and leaves all other days at 0 mm. For the wet days that have to be added, the simplest approach is to draw a straight line (green line in Fig. 4) between the lower point in the CDF, where wet days start in the observations (A in Fig. 4) to the point where the wet days start in the model at the corresponding precipitation value of the observations (B in Fig. 4). This approach was chosen by Themeßl et al. (2012).

However, the shape of precipitation in an ECDF is far from linear. Thus, this method only reproduces the wet day frequency of the observations but not the precipitation sums. In fact, it produces more precipitation than the observed precipitation. However, this method can deliver satisfactory results if the model bias in wet days is relatively low, e.g. well below 0.1 (i.e. 10 % of all days) in the ECDF.

To preserve the precipitation sum of the observations, wet days can be added by piecewise linear interpolation with the

constraint that the precipitation sum has to match the observations in the historical period. As this method should be applicable in future as well, the missing precipitation sum is defined as a relative quantity, i.e. the ratio of the missing precipitation in the lower part of the CDF and the whole precipitation (observational precipitation above 0.85 in the CDF in Fig. 4). This relative quantity can also be used for future time periods, as the precipitation sum of the upper part of the CDF can be calculated from the model after bias correction with EPPM. The algorithm to add wet days should be done between steps (2) and step (3) during

the EPPM method in Sect. 3.1:

Step (1): The number of missing wet days is calculated. This number can also be expressed as the difference on the y axis in the ECDF plot between Point A and B (Fig. 4).

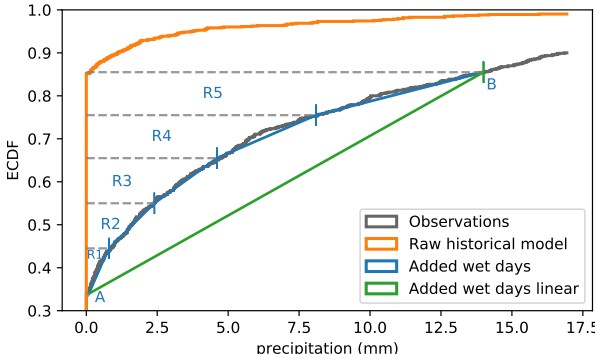

**Figure 4.** Schematic of two different methods to add wet days in the ECDF of precipitation. Themeßl et al. (2012) presented a simple linear method (green line) that leads to too high precipitation sums and is only suitable if the wet day bias of the model is low (below 10 % of total days). A more sophisticated method (blue lines) leads to very accurate precipitation sums that match the observations. The horizontal dashed gray lines are equidistant in y-direction and indicate the trapezoids, which are used for the calculation of piecewise precipitation sums.

Step (2): The precipitation sum in the observations between Point A and B in Fig. 4 as ratio to the whole observed precipitation is calculated.

Step (3): If the number of missing wet days exceeds a certain threshold, the Sect. between A and B is separated into equally sized segments on the y axis (Fig. 4). In this work, the threshold is defined as approximately 0.09 on the y axis in an ECDF, i.e. more than 9 % of all days are wet days in the observations that are dry days in the raw model. The number of segments is free to choose, but five parts are found to be a reasonable compromise between computational load and accuracy. For each of those (five) segments, the precipitation sum as ratio to the total sum in the observations is calculated.

Step (4a): After the EPPM bias correction, wet days are added. The number of wet days is taken from step (1). If they are below the set threshold of step (3), the simple method is applied: The precipitation values are linearly interpolated between the lower threshold (usually $0.1\,\mathrm{mm}$) and the lowest value of the bias corrected model data (in the historical period point B in Fig. 4)

Step (4b): If the distance of point A to point B exceeds the threshold as defined in step (3) the adding of wet days is done in 255 (e.g. five) segments. The precipitation sum of each segment can be expressed as the area of a trapezoid:

$$R = \frac{(u_{\mathrm{lower}} + u_{\mathrm{upper}}) \cdot \Delta y}{2} \cdot \mathrm{ndays} \tag{3}$$

where $R$ is the precipitation sum of each segment (R1, R2,...) which is the ratio calculated in step (3) multiplied with the total precipitation sum. $u_{\mathrm{lower}}$ ($u_{\mathrm{upper}}$) is the precipitation value at the lower (upper) end of the trapezoid. The height of the





trapezoid is $\Delta y$, ndays is the total amount of days. The calculation is done from bottom to top. In the example of Fig. 4 the

first trapezoid is:

$$R1 = \frac{(0.1 + u_{\text{upper}}) \cdot 0.1}{2} \cdot 900 \tag{4}$$

As $R1$ is known, this gives the first $u_{\text{upper}}$ . For the next segment $R2$ , the $u_{\text{upper}}$ from $R1$ turns to $u_{\text{lower}}$, where the next $u_{\text{upper}}$ is calculated. For the last segment $R5$, $u_{\text{upper}}$ is defined in a different way. As a smooth transition to the upper (bias corrected) part of the ECDF is preferable, $u_{\text{upper}}$ in $R5$ is defined as the lowest precipitation value from the bias corrected data (B in Fig.

265  4)

Step (5): The precipitation values for each segment are found by linear values between $u_{\text{lower}}$ and $u_{\text{upper}}$. Those wet days are then randomly assigned to dry days in the bias corrected model.

Step (6): For any future time period, wet days are added with the number of missing wet days found in step (1) and the missing amount of precipitation found in step (3).

## 3.3  Precipitation: conserving the CCS

EPPM (section 3.1) conserves the raw model's CCS on each quantile. For additive EPPM (used for temperature and dew point), this is also valid for means and sums. However, relative EPPM does not conserve the relative CCS of means for precipitation. As the precipitation sum (monthly sum, annual sum) is usually more important than the precipitation at a specific percentile in the CDF, an additional algorithm is developed to reproduce the raw model's change in means. This is referred to as the

conversation of the model CCS. Depending on the application of the corrected precipitation data, one can adjust the monthly or the annual CCS.

For a future time period, the CCS for precipitation for the raw model for one grid point is

$$\text{CCS}_{\text{mod}} = \frac{\overline{\text{R}_{\text{mod fut}}}}{\overline{\text{R}_{\text{mod hist}}}} \tag{5}$$

where $\overline{\text{R}_{\text{mod fut}}}$ is the mean precipitation of the model in the future time period and $\overline{\text{R}_{\text{mod hist}}}$ is the mean precipitation of

the model in the historical time period. The mean is either a monthly or annual climatological mean. The CCS for the bias corrected data after EPPM is

$$\text{CCS}_{\text{corr}} = \frac{\overline{\text{R}_{\text{corr fut}}}}{\overline{\text{R}_{\text{corr hist}}}} \tag{6}$$

The error $E$ of the CCS of the corrected model compared to the CCS of the raw model (in %) is defined as

$$E = \frac{\text{CCS}_{\text{corr}}}{\text{CCS}_{\text{mod}}} \cdot 100 - 100, \tag{7}$$





where a value of 0 is a perfect bias correction method. The precipitation (daily data) of the bias corrected model data $R_{\text{corr fut, t}}$ for every day t is adjusted with

$$R_{\text{corr CCS, t}} = R_{\text{corr fut, t}} \cdot \frac{CCS_{\text{mod}}}{CCS_{\text{corr}}} \tag{8}$$

to match the CCS of the raw model data. Equations 5 and 6 can be applied for either monthly or annual data or for both, applied one after the other. However, monthly and annual CCS can not be exactly conserved at the same time, because the
second CCS (e.g. the annual one) alters the data from the first CCS correction (e.g. monthly).

### 3.4    Scaled distribution mapping (SDM) and quantile mapping (QM)

To compare the performance of our new method EPPM, two other methods are also applied on the data. One of them is the traditional QM in a non-parametric form, which is widely used (e.g. Piani et al., 2010; Themeßl et al., 2011). The other one is the recently introduced parametric SDM that outperforms even improved methods of QM (Switanek et al., 2017). Both the
SDM and QM codes are available for via the pyCAT module.

As SDM is a parametric method, it fits functions to CDFs. For precipitation, gamma distribution can be selected. The parameters for the gamma distribution are found iteratively via the maximum likelihood function. Tests showed that the fitting is sometimes defective. The bias of the corrected precipitation data is still around 4 % on average and exceeds 50 % for single grid cells for the annual sum (Fig. 2). Hence, the SDM script is not always able to reproduce the past climate by correcting the
model according to the observations.

Therefore, we generated several versions of SDM. For our work, we improved the fitting of the gamma functions by adding initial guesses to the fitting function. According to the methods of moments (Thom, 1958; Wiens et al., 2003), the initial guess for the scale parameter $\theta$ for the gamma distribution is defined as:

$$\theta = \frac{\text{Var}(X)}{\bar{X}} \tag{9}$$

where $X$ is the data to be fitted and $\bar{X}$ is the mean of the data. Optionally also the shape parameter $k$ can be used for the initial guess as

$$k = \frac{\bar{X}^2}{\text{Var}(X)}. \tag{10}$$

We used four different versions of SDM which are as follows:

**SDM(raw):** This is the version of SDM as presented in Switanek et al. (2017). SDM(raw) lacks the correction of wet days, if
the model has too little wet days.

**SDM(0):** In addition to SDM(raw) corrected wet days are interpolated to the expected number of wet days which corrects a wet day bias. This algorithm was provided by the authors of Switanek et al. (2017).





**SDM(1):** In addition to SDM(0), the shape parameter $k$ is used as an initial guess for the gamma distribution of precipitation.

**SDM(2):** In addition to SDM(0), both shape parameter $k$ and scale parameter $\theta$ are used in the initial guess.

## 4 Results

EPPM is compared with QM and SDM with a focus on the three conditions expressed at the end of Sect. 1.

### 4.1 Conservation of historical climate

The four versions of SDM are compared with non-parametric QM and EPPMd to show the biases that are introduced by the bias correction methods themselves. We already showed the biases in the ÖKS15 and STARC-Impact data (Fig. 2). To
reproduce some of the biases, we used the smoothed observational data is produced in Sect. 2. Depending on the method of bias correction, even after correction the bias can be significant (Fig. 5). Figure 5a is the average annual precipitation as in OBS, Fig. 5b is the artificial smoothed model. When comparing area means of (a) and (b), the difference is negligible. The only difference is the spatial resolution between OBS and the model. Figure 5c uses SDM(raw) which produces the largest errors. The difference in annual precipitation exceeds $100\,\mathrm{mm}$ in parts of East Tyrol and Carinthia (southwest parts of Austria).
The errors produced with SDM(0) and SDM(1) (5d and e) are considerably smaller. The smallest errors are produced when using SDM(2) where the error is $10\,\mathrm{mm}$ or less in most parts of Austria. For comparison, the model data was corrected with other methods like QM where the error is close to zero. This is because QM corrects with exact empirical CDFs of both model and OBS which produces very accurate results in the reference period. QM outperforms all other methods. The remaining error of our method EPPMd originates from the empirical CDF which is defined on 100 discrete values. Therefore, the most extreme
values can not be accurately reproduced which leads to small errors.

### 4.2 Precipitation: performance on dry models

As already discussed in Fig. 5, parametric methods do not always reproduce the observational climate. Furthermore, very few bias correction methods accurately bias correct climate models with a distinct dry bias. To show the skills of EPPMd, we compare SDM, QM and EPPMd using the introduced model with a large dry bias in the order of hundreds of mm precipitation
per year (Fig. 6a and b). The difference of the model corrected with SDM(2) minus OBS shows feasible results, but overall the corrected data shows a slight wet bias that exceeds $25\,\mathrm{mm}$ at some grid cells (Fig. 6c). The area mean annual bias of SDM(2) is 6 mm. QM corrects the precipitation at already existing wet days but cannot add wet days. Thus, there is still a dry bias after bias correction (Fig. 6d) where the area mean is -48.5 mm. EPPM shows a similar pattern (Fig. 6e) as both methods is almost identical in the historical period with the only difference that EPPM uses 100 discrete percentiles and QM uses all values for
the CDFs. EPPMd adds wet days and is able to accurately reproduce climatological precipitation sums in the historical period where the average annual precipitation bias is only 2.5 mm.

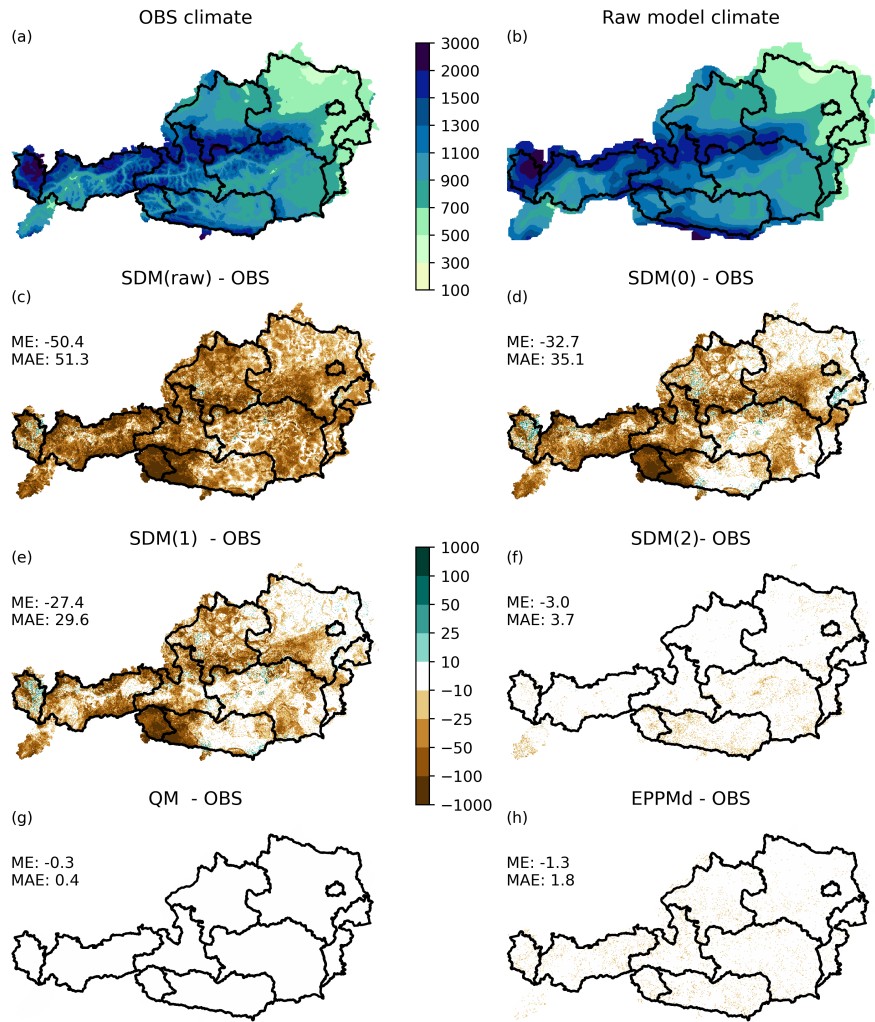

**Figure 5.** Bias correction of precipitation data. The model is produced by smoothing OBS. (a): Observational annual precipitation. (b): Raw model annual precipitation. (c)-(h): Difference in annual precipitation (model minus observational data) in mm for (c) SDM(raw), (d) SDM(0), (e) SDM(1), (f) SDM(2), (g) QM and (h) EPPMd. ME: mean error. MAE: mean absolute error.

When it comes to wet days, SDM(2) tends to produce too many new precipitation days (Figure 7c). The average annual day bias is 15.5 days. The reason for this positive wet bias is still in discussion We assume that it can be caused by the fitting of gamma functions to the CDFs which introduces new errors. Both the QM and EPPM (Figure 7d and e) cannot change the number of wet days, so the average annual wet day bias of -69.9 days of the raw model is unchanged. EPPMd (Figure 7f) performs best of all methods with an average bias of only -2.3 wet days per year. Only very few grid cells exceed a wet day bias of +10 or -10 days.





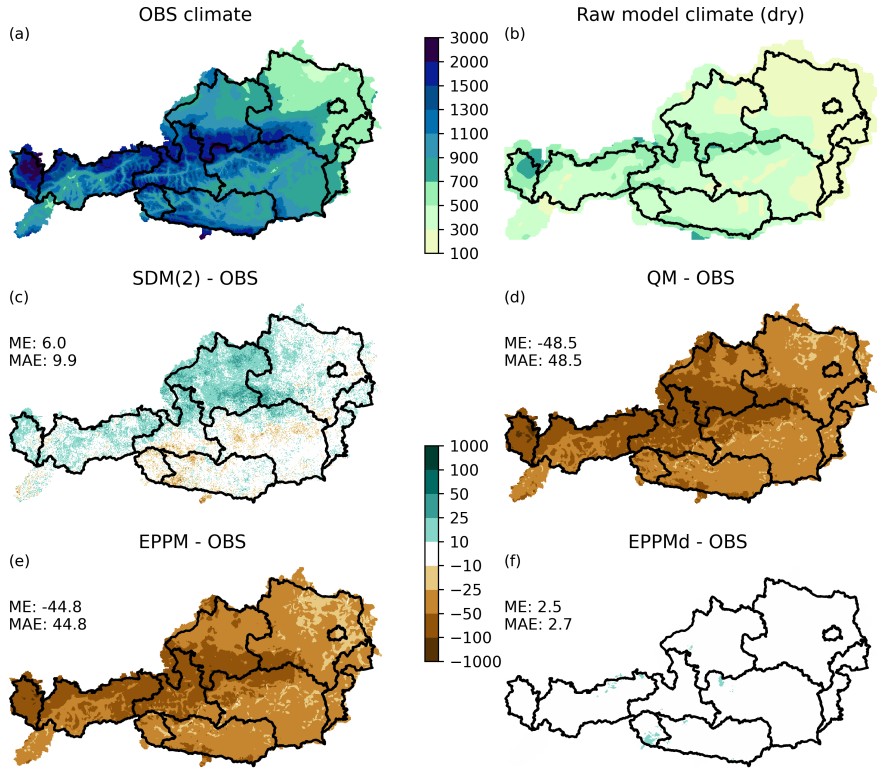

**Figure 6.** Climatological annual precipitation (mm) sum in the historical period for dry model data. (a): OBS annual precipitation. (b): Raw model (dry) annual precipitation. (c)-(f): Difference in annual precipitation (model minus observed data) in mm for (c) SDM, (d) QM, (e) EPPM and (f) EPPMd.

Fig. 8 shows the correction of the dry model with SDM and EPPMd on a single grid cell. In the observational data set (black line), only 30 % of the days are without precipitation. In the raw model, more than 55 % are dry days (orange). In the future 350 period, there are even more dry days with almost 70 % or all days (red). EPPMd corrects the historical data so that is matches the CDF of the observations (blue). For the future period, EPPMd adds wet days according to the absolute difference of wet days of observations and historical raw model (Fig. 8a). The shape of the EPPMd CDF is caused by the unsteadiness of the raw future model and are more prominent due to the multiplication with CVs much larger than 1. In comparison, the data corrected with SDM(2) slightly deviates from the observational data in the historical period. In the future period, SDM(2) is smoother 355 than EPPMd but the number of wet days is significantly higher which is caused by different definitions of the wet day bias. SDM uses the ratio of wet days between model and observations in the historical period, whereas EPPMd uses an absolute difference of wet days.



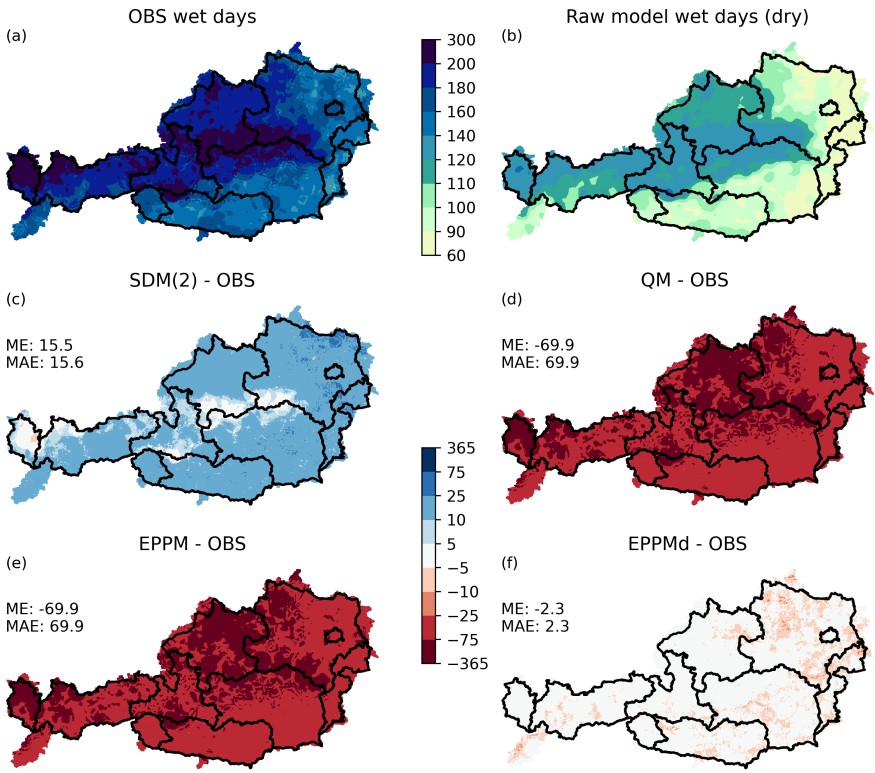

**Figure 7.** Wet days per year ($\geqslant 0.1$ mm) in the historical period for dry model data. (a): Annual wet days in OBS. (b): Raw model annual wet days. (c)-(f): Difference in annual wet days (model minus observed data) for (c) SDM, (d) QM, (e) EPPM and (f) EPPMd.

## 4.3 Climate change signal (CCS)

One of the three main conditions for EPPM is that the CCS of the raw climate model should not be altered. As stated by Maraun
(2016) methods like standard QM can add a systematical error to the temperature CCS where the CCS is defined as an absolute value. Therefore, we show that EPPM automatically conserves the temperature CCS. As temperature data, we used the artificial data as produced in Sect. 3.1. The corresponding CDFs are shown in Fig 3. Fig. 9 shows the increase of temperature in the raw model of 0.42 °C each decade. Three bias correction methods (SDM, QM and EPPM) are compared. SDM and EPPM are able to exactly conserve the CCS of the climate model which is a linear trend of 0.42 °C per decade. In contrast, QM inflates the
climate change signal with a linear trend of 0.71 °C per decade.

For precipitation, the CCS is defined as a relative one, i.e. the precipitation mean of the future climate divided by the precipitation in the historical climate. The relative CCS is greater than 1 in case of more precipitation of future. Figure 8 shows a single grid cell with the CDFs for the observations and a raw model (with less wet days and less precipitation sum that the observations). The model data was corrected with EPPMd and SDM. In the second line on top of Fig. 8 a ratio measure for the
conservation of the CCS is shown. It is the ratio between the CCS of the corrected model and the CCS of the raw model. A

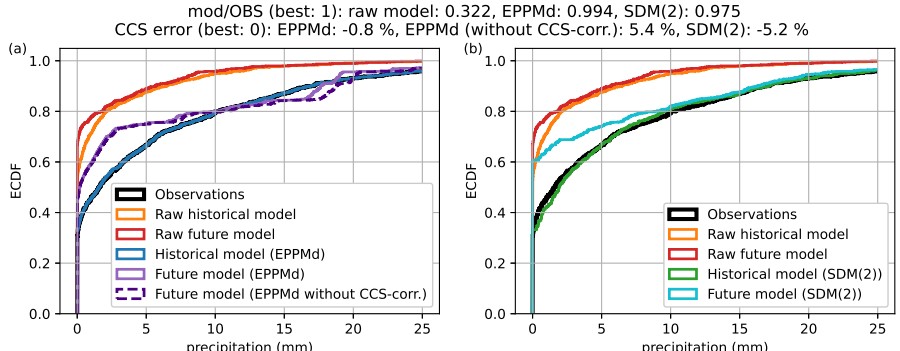

**Figure 8.** Bias correction of precipitation data for a dry model with less wet days than the observations. Both panels: CDF of observations (black), raw model data for the historical time period (orange) and the future time period (red). Left: Corrected model with EPPM (blue and purple) and EPPMd (dashed purple). Right: Corrected model with SDM(2). First line on top: ratio of precipitation means from historical model and observations. A perfect model equals a ratio of 1, i.e. the precipitation sums in model and observations are equal. Second line on top: Error of CCS (Eq. 7). A perfect bias correction equals 0 %.

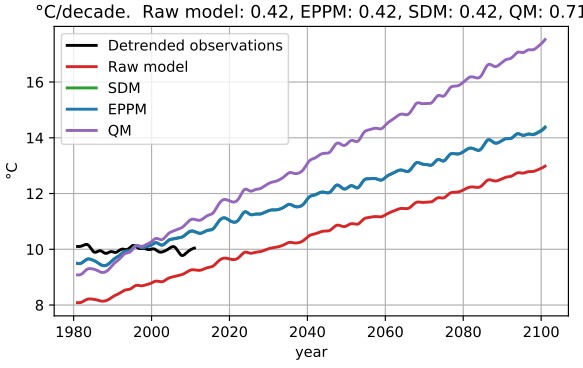

**Figure 9.** Running means of temperature data of detrended observations, the raw model and 3 different bias correcting methods (QM, SDM and EPPM). SDM and EPPM are almost identical. On top: linear trends in °C per decade for each bias correction method. The data is identical to the data used in Fig. 3.

value close to 1 means that the bias correction method is able to conserve the CCS. For the data shown, SDM(2) and EPPMd without the algorithm for correcting the CCS (see Sect. 3.3) perform almost equally with a ratio of 0.948 respectively 1.054. When using the CCS correction after EPPMd, the best result is of all methods is achieved with a ratio of 0.992.

In Sect. 2 an artificial dry model was produced by drying OBS. Figure 10 shows again the ratio of the CCS of corrected model data and raw model data for the dry model. As before, SDM generally underestimates the CCS while EPPMd (without the CCS correction) and QM generally overestimates the CCS. The overall bias of EPPMd is lower than the bias of SDM and





QM. In Fig. 10d with EPPMd the CCS of the annual precipitation is forced to match the raw model CCS, therefore the ratio is exactly 1.

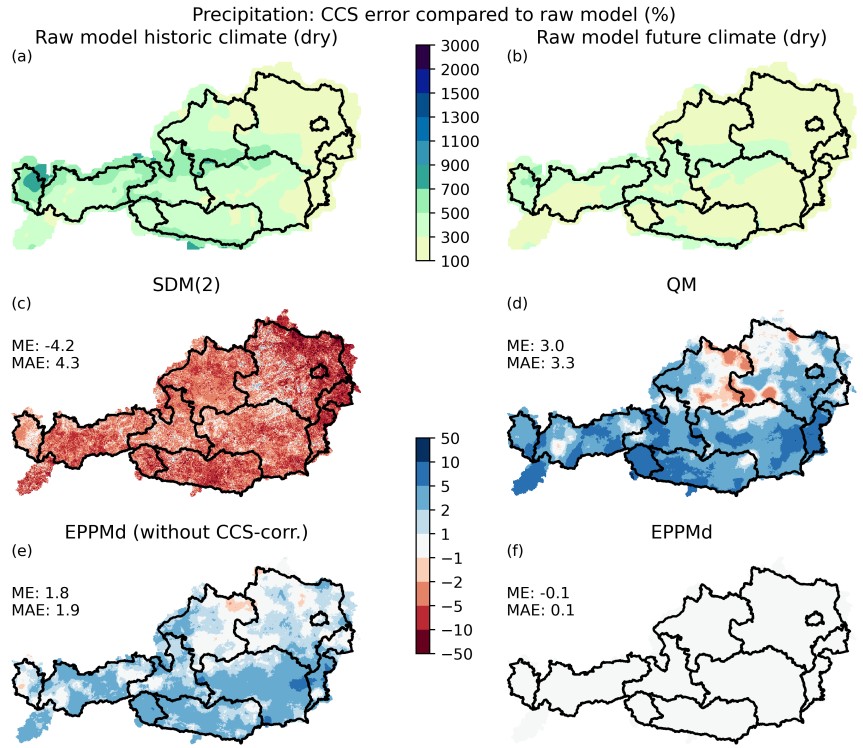

**Figure 10.** Error of CCS compared to CCS of raw model (Eq. 7). (a): Raw model annual precipitation in historic period (mm). (b): Raw model annual precipitation in future period (mm). (c) SDM, (d) QM, (e) EPPMd without additional correction of the CCS and (f) EPPMd. A perfect bias correction equals 0 %.

## 5  Conclusions

Statistical bias correction methods are widely used to improve direct model output from climate models but can not fully remove all model errors. The corrected data is often used as input for climate impact studies where one has to be aware of the limitations of the method. We introduced the new method EPPM that solves some of the known errors. EPPM statistically corrects the output of climate models via adjusting the CDF of different model output variables (such as temperature and precipitation) to match the observation's CDF. Unlike most other existing methods this method merely corrects at constant percentiles between model and observation, also in future time periods. EPPM corrects models that have large trends over time (CCS) and conserves the CCS of the raw model. When correcting the model with EPPM, it is assumed that the model bias stays constant over time rather at certain percentiles than at specific absolute values of a meteorological variable (unlike traditional



quantile mapping). Precipitation is particularly challenging to correct because the magnitude as well as the frequency of wet days has to be corrected. An additional algorithm after EPPM (called EPPMd) takes account of dry day frequency biases by adding wet days. EPPMd performs at least as good as other methods like QM or SDM, but exclusively unites the following features simultaneously:

- It hardly enhances or suppresses the CCS in contrast to traditional QM, i.e. EPPM explicitly reproduces the same CCS as in the raw model. For additive EPPM (e.g. for temperature), this is valid without any limitation. If the correction factors in EPPM are applied multiplicatively (e.g. for precipitation), the CCS is defined as a relative one. In general, the relative CCSs of monthly and annual means differ from the relative trends at percentiles. Depending on the application one has to decide whether to conserve the relative CCS at percentiles or at means. For the former one, an algorithm adjusts after EPPM adjusts monthly or annual means to force the raw model's CCS.

- EPPM is capable of statistically correcting the model's past climate to fit the observations accurately. This is mostly due to the fact that it is a nonparametric method, i.e. it uses empirical distribution functions instead of fitted functions (for variables like temperature, precipitation etc.) which allows the CDF to follow any possible shape. The fitting of functions will always produces errors which can be reduced to a minimum with a good fitting algorithm but can never be fully removed.

- It preserves precipitation sums and wet days. If the model has too little wet days, an subsequent algorithm adds additional wet days in order to reproduce the observation's precipitation sums and wet day frequency. If the model has too many wet days, the wet day frequency is automatically corrected with EPPM.

Table 1 summarizes our main results on these three items. We assume that SDM can be seen as a representative of parametric methods in general because the errors introduced with SDM are mainly due to the fitting of functions.

A good performance of the corrected data in any of those three conditions is crucial, as it is used as input for further impact studies. Impact models (e.g. plant growth models) are often calibrated with bias corrected historical meteorological data from a climate model. The focus of impact studies often lies on the CCS. If an impact model is calibrated with inaccurate meteorological data in the past, the impact of climate change can lead to wrong conclusions even if the CCS itself is accurate.

Compared to methods that fit functions for the CDFs of the variables (parametric methods), EPPM is computationally inexpensive and can therefore be used for high resolutions and/or large domains. For instance, EPPM is 1 to 2 orders of magnitude faster than SDM. This is of particular importance because data users often need high resolution data as input for further studies (e.g. Seidl et al., 2019).

EPPM corrects each pixel and each variable independently from others and therefore belongs to the group of univariate bias correction algorithms. Like other univariate algorithms, it does not improve spatiotemporal statistics, i.e. the distribution of precipitation fields on specific days in the model or the persistency of weather patterns (Pastén-Zapata et al., 2020; Potter et al., 2020; Charles et al., 2020). Multivariate methods have already been introduced but suffer from disadvantages such as very high computational demands or a limited measure of the full multivariate dependence of structure (e.g. Cannon, 2018; Bürger



**Table 1.** Qualitative comparison of bias correction methods for three conditions: (1) The conservation of the CCS (either absolute or relative) compared to the raw model. The relative CCS (e.g. for precipitation) can either be conserved at percentiles or at (monthly, seasonal or annual) means (see Sect. 3.3). (2) The capability of reproducing the historical climate of the observations and (3) the capability of correcting models with less wet days than the observations.

| Method | Absolute CCS (temperature) | Relative CCS (precipitation) | Historical Climate | Models with less wet days than observations |
|---|---|---|---|---|
| **SDM(raw)** | yes | yes (only at percentiles) | no | no |
| **SDM(0)** | yes | yes (only at percentiles) | no | partially |
| **SDM(2)** | yes | yes (only at percentiles) | partially | partially |
| **QM** | no | no | yes | no |
| **EPPM** | yes | yes | yes | no |
| **EPPMd** | yes | yes | yes | yes |

et al., 2011). For more realistic times series, a two-step approach is introduced by Volosciuk et al. (2017) that consists of QM on the model's spatial resolution in a first step and a downscaling with a stochastic regression-based model as a second step which adds random small-scale variability. However, the added skill varies and may even increase the bias in certain cases. Also, every statistical bias correction comes to its limits if the performance of the model in consideration is poor (garbage in - 
garbage out).

*Competing interests.*   The authors declare that they have no conflict of interest

*Acknowledgements.*   We acknowledge the precipitation data set from SPARTACUS by ZAMG. We also thank Copernicus Land Monitoring Service as part of the European Environment Agency (EEA) for the topography data.





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
