# Peer review of "An improved statistical bias correction method that also corrects dry climate models"

_Hydrology and Earth System Sciences, 2020_

## Short Comment (SC1) · 29 Oct 2020

Jorn Van de Velde

jorn.vandevelde@ugent.be

As the changes to the climate change signal induced by quantile mapping and other bias adjustment methods are still a matter of debate, this paper could be a worthy addition to the literature.

However, I would like to ask if the authors are aware of the paper by Vrac et al. (2016). This paper builds on Themeáždl et al. (2012) by introducing the SSR method, which is a flexible precipitation occurrence-bias-adjusting method that can also correct a negative wet day bias. This is the only advanced method deliberately constructed to adjust models with too few wet days, yet this is not discussed in the paper. It would be interesting to read at least a discussion on the differences between this method (based

on, but not limited to, the CDF-t method) and the one proposed in the paper, or maybe even see a quantitative comparison.

Reference:

Vrac, M., Noël, T., and Vautard, R.: Bias correction of precipitation through Singularity Stochastic Removal: Because occurrences matter, Journal of Geophysical Research: Atmospheres, 121, 5237–5258, https://doi.org/10.1002/2015JD024511, 2016
* * *

---

## Author Comment (AC1) · 2 Nov 2020

Thank you very much for your comment. We will definitely consider the paper by Vrac et al. (2016) in our paper. The introduced SSR method seems to be a simple yet effective method to correct climate models with too many dry days. Quick quantitative comparisons with our method show that the SSR method performs almost as good or even similar as our own method.

However, the bias correction method used by Vrac et al. (2016) is CDF-t which does not conserve the climate change signal as indicated in the Fig. 11, 12 and 13. Further comparisons on the climate change signal after bias correction with CDF-t are shown in the paper of Pierce et al. (2015) which you can find in our reference list. They explicitly

state that CDF-t may alter the temperature trends found in the raw climate model.

Therefore, the method introduced by Vrac et al. (2016) is a valuable approach for the correction of the wet day frequency. Concerning the bias correction method, there are already other, improved methods in the literature, including our suggested EPPM.

———————————————

---

## Referee Comment (RC1) · Anonymous Referee #1 · 11 Dec 2020

Firstly, I felt the title of the paper could be improved. The authors are not correcting dry climate models as that requires new rainfall parameterisations. It would be worth thinking carefully what the contributions are and then reflect them in the title. Secondly, I feel the approach presented has significant similarities to quantile delta mapping (QDM) of Cannon et al. (2015) and equidistant CDF matching method (EDCDFm) of Li et al. (2010) as well as to another variant of quantile mapping in the details below. Authors need to really focus on distinguishing their approach against the others, maybe using contrived synthetic examples where advantages can be highlighted, or using real data along the lines they have already done. Furthermore, I am still left with doubts on whether I have understood their approach, as their presentation is not mathematical enough for the reader to be confident. Some work on this aspect is needed.

[Figure]

My detailed comments are (in the order I read the paper): Line 35: I feel the authors need to also acknowledge the papers on correcting systematic biases that have been written with hydrological systems in mind. I am referring to those studies that attempt to correct biases in persistence, which is critical when a sequence (time-series) of inputs are coming from a GCM to drive a hydrological simulation. Some examples are 10.1016/j.jhydrol.2016.04.018, 10.1007/s00382-016-3510-z, 10.1029/2018WR023270, and these are by no means exhaustive so authors should look into other papers as well that have been written with a hydrological application in mind. Line 94: pretty sure both wet and dry day biases are corrected in MBC - 10.1016/j.envsoft.2018.02.010 by resampling additional wet days. Line 110: The statement of no extrapolation is needed needs to be clarified to be in line with the statement in line 180, for the extreme values. Line 159, 185: Is there any consideration for subtracting a linear trend to detrend the time series (Step (1) and Step (6)) rather than any non-linear trend which may contain in the time series? Line 168: What is/are the parameter(s) for the trial and error procedure to be said satisfactorily provide 100 correction values? And how is it relevant with the statement in line 329 which states that the remaining error of the proposed approach due to the defined-100 discrete values? Line 171 and 175: The correction value formulas which are presented are for the simulated-current time series. Will it be applicable to bias correct the simulated-future time series? Should the bias correction procedure of the simulated-future time series following the bias correction procedure of the simulated-current time series? I am just not sure what this does to the definition of a CDF, as I can see cases where the corrected value exceeds a limit of 1. Perhaps the authors have a good reason for adopting this approach, but it has not been motivated well enough I feel. Also, authors need to improve the way they present equations. CV in equation 1 is a function of a CDF of (I think) the observed series. Where this correction is applied is not clear to me if the CV is indexed with respect to the observed series. Is it applied to future simulations? Does that not create an inconsistency? I think this part is critical to the paper and it can be presented a lot better. Line 180: How can Step (5) be applicable for the simulated-future time series in which most of the data points have a higher magnitude compare to the simulated-current time series? Line 193: Step (9) is more like additional information given with respect to the detrending procedure in Step (1) and Step (6). Therefore, the order of the step might be revised appropriately to provide a clearer stepwise procedure. Line 196: I feel the authors may be presenting the same approach as 10.1029/2009GL038401. A more mathematical presentation of the approach is needed, along with a discussion of potential issues that are created. I feel, for one, the authors may be introducing a bias in their representation of zero rainfall, but more details are needed. Line 290: It is stated that the climate change signal cannot be conserved for multiple time scales using the proposed procedure, with the discussion implying it can be done for both. This is analogous to the nesting approach which has been used a lot in the bias correction literature 10.1175/jcli-d-15-0356.1. Figure 9. Will the proposed approach applicable to bias correct simulated-future time series which contain different trend with simulated-current time series?

---

## Referee Comment (RC2) · Anonymous Referee #2 · 12 Jan 2021

Review of

'An improved statistical bias correction method that also corrects dry climate models'

by F. Lehner, I. Nadem and H. Formayer

**Recommendation: reject and resubmit new manuscript**

This manuscript presents a bias correction method that preserves the simulated climate change signal (CCS) and allows to increase the number of dry days. The method is called 'Empirical percentile-percentile mapping' (EPPM). As stated by the authors, it uses elements of published methods to achieve the CSS preservation and the adding of dry days, and in this respect there is no novelty.

However, EPPM deviates strongly from common practice with respect to how the corrections are applied to future climate. Usually in quantile mapping (QM) the percentile for a given simulated value in simulations for future climate is determined with respect to the cumulative distribution function (CDF) of the simulated values in the present climate and then mapped onto the observed values for the same percentile. This means a simulated value is always mapped onto the same corrected value regardless of whether it occurs in the present or in the future climate. This assumption of constant bias for a given value does not necessarily always hold, but in lack of knowledge how the bias might change from the present to the future climate, it is a reasonable assumption. In contrast EPPM determines for future climates the percentile for a given simulated value from the future simulated CDF and applies the correction derived in the present climate for this percentile (the difference or ratio between the observed and simulated values for this percentile). This means the corrections applied to a given simulated value can be very different in the present and the future climate. I cannot see any physical justification for doing this. Unfortunately the manuscript neither explains this key property in a transparent way nor does it give any justification for this approach. Even the name of the method is misleading, because for the future climate there is no mapping of percentile values for one distribution onto those for another.

Moreover the text is written as if it was obvious that the CCS should be conserved by postprocessing methods, but there a reasonable arguments for and against this. Again, there is no justification at all for this in the manuscript and related literature is not sufficiently discussed.

In its current form the manuscript is conceptually unclear, not systematically written, contains only superficial explanations of the approach, and if published it would add confusion to the discussion on bias correction rather than help to address open issues and contribute to methodological progress. There might be aspects in this work that are publishable, but getting the manuscript in a publishable form requires a clear identification of what is novel, and a systematic and sound explanation and justification of the approach. This would go substantially beyond a revision of the manuscript and constitute a new paper. I thus recommend rejecting the manuscript, but encourage submission of a new manuscript after addressing the problems.

More detailed comments on the issues mentioned above and some additional points are listed below.

**Specific comments**

1)
The introduction cites a good number of relevant publications, but it is not well structured and many details are unclear. The attempt at grouping the methods in lines 67-81 is not clearly linked to the discussion of individual methods at the beginning of the introduction.

The introduction should be organised from the beginning by explaining and giving examples for groups of methods with common structural elements and properties, including

- Additive and multiplicative scaling, and linear regression
- Empirical QM, parametric QM using one function, multi-segment parametric QM
- Modification or preservation of the simulated CCS, specifying what aspect of the CCS is preserved, e.g. mean or CCS for specific quantiles
- Treatment of biases in wet-day frequency
- Treatment of values outside observed range
- Which methods assume stationarity of the bias? For those who don't, how do they specify potential non-stationarities?

It should also include a paragraph on comparison studies. This overview should include Maraun and Widmann (2018), which is a standard reference for downscaling and bias correction, and the VALUE comparison of downscaling and bias correction methods (Maraun et al. 2018, Gutierrez et al. 2019, Widmann et al. 2019).

Gutierrez, J.M., D. Maraun, M. Widmann et al., 2019: An intercomparison of a large ensemble of statistical downscaling methods over Europe: Results from the VALUE perfect predictor cross-validation experiment. *Int. J. Climatol*, **39**(9), 3750-3785.

Maraun, D., R. Benestad, S. Kotlarski, E. Hertig, M. Widmann, J. Wibig, J.M. Gutierrez, R. Huth, R.E. Chandler and R. Wilcke, 2019: Validation of temporal variability in the VALUE perfect predictor experiment. *Int. J. Climatol*., **39**(9), 3786—3818.

Maraun and Widmann, 2018: Statistical downscaling and bias correction in climate research. *Cambridge University Press*, ISBN 1107066050

Widmann, M, J. Bedia, J.M. Gutierrez, T. et al., 2019: Validation of spatial variability in downscaling results from the VALUE perfect predictor experiment. *Int. J. Climatol*., **39**(9), 3819-3845.

2)
The list of properties that the new method should satisfy (lines 97-100) is not clear. Property 1: What are 'long-term trends'? Differences between the simulated future and reference periods? If so, differences in the mean or in individual quantiles? Property 2: 'the model data' should be 'the corrected model data'. 'should match the observational data' with respect to what? Means, quantiles, trends?

3)
The outline of the method (lines 107-112) is unclear. Up to here the text has mentioned as key issues the correction of an underestimation of wet days, and the preservation of the CCS. Now the question of whether biases are stationary is the key point, but this and the various ways of how to define QM have not been properly introduced. Most QM methods assume stationarity, not because it is necessarily a correct assumption, but because deviations from this have to have specific forms, for instance the one given in this manuscript, which are difficult to justify. As mentioned at the beginning of the review any specific approach needs of course to be justified.

4)
Lines 102/103: the underestimation of the precipitation sums needs to be explained in more detail. If the full PDFs were matched the sums and thus the means would be identical by construction because $mean(x) = \int x \, pdf(x) \, dx$ . If I understand correctly the problem is that the PDFs are not matched for the full range of values including zero, but only for the simulated wet days, which leads to the mean over the corrected simulated wet days matching the observed sum over these days, but the simulated mean over the whole period includes more dry days than the observed mean over the whole period.

5)
Whether the CCS of the original simulations should be preserved by bias correction is an important questions, but the correct answer depends on whether the raw CCS can be expected to be realistic. Discussions of this can be found for instance in Maraun and Widmann (2018), section 12.9.1, and in Maraun et al. (2017). The paper contains no justification for preserving the CCS.

Maraun, D, T. Shepherd, M. Widmann, et al., 2017: Towards process-informed bias correction of climate change simulations. *Nature Climate Change,* **7**(11), 764-773.

6)
The validation of the OEKS15 data in section 4 is not convincing with the information available in the manuscript. In addition to structural elements in SDM that can lead to a difference in the means of the postprocessed models and the observations there are two more potential reasons for this difference. The first is the use of different observation datasets used for fitting SDM and for validation SDM. The second is a different reference period for fitting SDM and for the validation. There is no information in the manuscript on these two points, and therefore it cannot be concluded that the differences that are found are due to the structure of SDM.

---

## Author Comment (AC2) · 9 Feb 2021

**General remarks**

We thank the referees for their suggestions. We have restructured and redrafted the manuscript by addressing all concerns of both reviewers. The revised manuscript is now ready for a new upload as soon as the editor invites us to do so.

We now explicitly state that our method is based on existing methods but strictly applied non-parametric which is an important distinctive feature. Following the suggestions of referee #2, we changed the name of the method. The new name is eQC (Empirical quantile correction). We postulate three properties and discuss them in the introduction. We test which quantile-based bias correction methods are able to fulfil them and have now restructured the whole manuscript according to those three properties.

**Reply to referee #1**

Thank you very much for your review. We want to reply to your comments one by one:

- *Firstly, I felt the title of the paper could be improved. The authors are not correcting dry climate models as that requires new rainfall parameterisations. It would be worth thinking carefully what the contributions are and then reflect them in the title.*
  We agree with you that the title could be improved and maybe also be more concise. Since the outline of the manuscript has now changed from the presentation of a new method to an evaluation and a combining of existing methods (see below), the title will be changed to "Evaluating quantile-based bias correction methods"

- *Secondly, I feel the approach presented has significant similarities to quantile delta mapping (QDM) of Cannon et al. (2015) and equidistant CDF matching method (EDCDFm) of Li et al. (2010) as well as to another variant of quantile mapping in the details below. Authors need to really focus on distinguishing their approach against the others…*
  After rechecking the literature about EDCDFm and quantile delta mapping (QDM) we concluded that our method is very much similar to both of them. Equation 2 in Li et al. (2010) is the correct mathematical description for our method when correcting temperature. For precipitation, our method is almost equivalent to PresRAT (Pierce et al., 2015). The important difference to these methods is that our method is strictly empiric/nonparametric. Therefore it is necessary to use a new name and we chose eQC (empirical Quantile Correction).
  Our paper focuses on the application and further improvement of existing bias correction methods especially for precipitation along with some new ideas already introduced by Pierce et al (2015), Cannon et al. (2015) and Vrac et al. (2016). We compare our method to SDM (Switanek et al., 2017), a method which is very much similar to QDM. SDM is parametric (it fits functions to the CDFs), whereas for QDM it is not defined weather to use a parametric or non-parametric approach.
  We showed that under certain conditions parametric approaches like SDM come with disadvantages such as resource heavy computations and errors even after bias correction, especially when comparing the means of historical and model climate.

- *… maybe using contrived synthetic examples where advantages can be highlighted, or using real data along the lines they have already done.*
  For comparisons we used contrived synthetic examples (Fig. 3, Fig. 4, Fig. 9 – now Fig. 6) and real observations along with contrived synthetic model data. We avoided to use real model data to show that the errors of some bias correction methods arise merely from the method itself and not from bad model data.

- *Furthermore, I am still left with doubts on whether I have understood their approach, as their presentation is not mathematical enough for the reader to be confident. Some work on this aspect is needed.*

Mathematical equations are now improved and clarified. For easier comparisons we use a similar nomenclature as in Li et al. (2010).

- Line 35: *I feel the authors need to also acknowledge the papers on correcting systematic biases that have been written with hydrological systems in mind. I am referring to those studies that attempt to correct biases in persistence, which is critical when a sequence (time-series) of inputs are coming from a GCM to drive a hydrological simulation. Some examples are 10.1016/j.jhydrol.2016.04.018, 10.1007/s00382-016-3510-z, 10.1029/2018WR023270, and these are by no means exhaustive so authors should look into other papers as well that have been written with a hydrological application in mind.*
  We appreciate your suggestions on additional references that focus on bias correction with hydrological systems in mind. The new bias correction approaches of those papers focus on slightly other aspects as temporal autocorrelation and spatial crosscorrelation whereas our approach focuses on precipitation mean and wet days. They are very valuable contributions to bias correction and are happy to include them to our references.

- Line 94: pretty sure both wet and dry day biases are corrected in MBC - 10.1016/j.envsoft.2018.02.010 by resampling additional wet days
  The correction of dry and wet day biases is not new to bias correction. Your lead to the approaches by e.g. Cannon et al. (2015) will be included in our revisions.

- Line 110: *The statement of no extrapolation is needed needs to be clarified to be in line with the statement in line 180, for the extreme values.*
  We use empirical CDFs with 100 values. Therefore, some extrapolation or interpolation is necessary to calculate values in between, above or below the discreet 100 CDF values. For our method, we used linear interpolation and constant extrapolation of the correction values (sometimes referred to as transfer function). Our statement that no extrapolation is needed is confusing. We will delete this statement. What we wanted to point out was that in contrast to traditional QM less extrapolation is needed. This is because the occurrence of new extremes does not increase the amount of extrapolation that is needed. In our case the extrapolation is necessary for 0.5 % on both end of the distribution.

- Line 159, 185: *Is there any consideration for subtracting a linear trend to detrend the time series (Step (1) and Step (6)) rather than any non-linear trend which may contain in the time series?*
  We used a linear trend to detrend the time series, as this is a common approach and was also done in Switanek et al. (2017). Other trends like polynomials of higher order may also be possible but this was not the focus of our research.

- Line 168: *What is/are the parameter(s) for the trial and error procedure to be said satisfactorily provide 100 correction values? And how is it relevant with the statement in line 329 which states that the remaining error of the proposed approach due to the defined-100 discrete values?*

The sampling of 100 points seems to be a reasonable compromise for usually approximately 900 data values. A higher amount would be less robust to extremes, as especially the CVs of extremes would depend even more on single extreme events. A lower number provides less details about the distributional shape of the model bias.

- Lines 171, 175, 180: *The correction value formulas which are presented are for the simulated-current time series. Will it be applicable to bias correct the simulated future time series? Should the bias correction procedure of the simulated-future time series following the bias correction procedure of the simulated-current time series? I am just not sure what this does to the definition of a CDF, as I can see cases where the corrected value exceeds a limit of 1. Perhaps the authors have a good reason for adopting this approach, but it has not been motivated well enough I feel. Also, authors need to improve the way they present equations. CV in equation 1 is a function of a CDF of (I think) the observed series. Where this correction is applied is not clear to me if the CV is indexed with respect to the observed series. Is it applied to future simulations? Does that not create an inconsistency? I think this part is critical to the paper and it can be presented a lot better. Line 180: How can Step (5) be applicable for the simulated-future time series in which most of the data points have a higher magnitude compare to the simulated-current time series?*
We are improving the mathematical writing which should answer your questions in this regard. Also, our references to Li et al. (2010) and Pierce et al. (2015) should answer some of those questions. We assume that most of your concerns arise from the unclear mathematical equations.

- Line 193: *Step (9) is more like additional information given with respect to the detrending procedure in Step (1) and Step (6). Therefore, the order of the step might be revised appropriately to provide a clearer stepwise procedure.*
The stepwise procedure is now completely restructured after step (4).

- Line 196: *I feel the authors may be presenting the same approach as 10.1029/2009GL038401. A more mathematical presentation of the approach is needed, along with a discussion of potential issues that are created. I feel, for one, the authors may be introducing a bias in their representation of zero rainfall, but more details are needed.*
The CDF-t approach by Michelangeli et al. (2009) is a different one. This is graphically shown in Pierce et al. (2015). Our method is mathematically the same as EDCDFm and PresRAT with the difference that it is purely nonparametric. The number of wet/dry days is corrected with very high accuracy as shown with (extreme and artificial) model data in Fig. 7 (now Fig. 9) in our paper.

- Line 290: *It is stated that the climate change signal cannot be conserved for multiple time scales using the proposed procedure, with the discussion implying it can be done for both. This is analogous to the nesting approach which has been used a lot in the bias correction literature 10.1175/jcli-d-15-0356.1*
In the revised version, we now mention nesting approaches. But it seems to us the

nesting approach (if using some form of multiplicative quantile mapping like we did) cannot conserve the change of means. It conserves the change of medians across selected time scales. Our approach to correct the CCS is analogous to the one e.g. in Pierce et al. (2015).

- Fig. 9 (now Fig. 6): *Will the proposed approach applicable to bias correct simulated-future time series which contain different trend with simulated-current time series?* We tested three methods with different trends. For example, we tested a sinusoidal shape of the trend with rising temperatures in the beginning, a peak around 2020 and decreasing temperatures during the later 21th century. For clarity, we decided to show the linear trend, where the deficiencies of QM can be clearly seen.

[Figure]

**Reply to referee #2**

**General reply**

Thank you very much for all your suggestions. We have restructured and redrafted the manuscript by addressing all your concerns. We do not see the need for a completely new submission, as the structure of the manuscript is changed in the following way where we compare quantile-based bias correction approaches. After rechecking the literature according to referee #1, our method is very similar to EDCDFm from Li et al. (2010) and was further developed by Pierce et al. (2015) to PresRAT. The difference to these methods is that our method is strictly empiric/nonparametric. So, we decided to use a new name. As there are already so many names out there, we chose the name eQC (empirical Quantile Correction) to emphasize the empirical correction of quantiles.

Also, there are already existing approaches for the other issues that we addressed. The monthly or annual correction of the climate change signal is shown in Pierce et al. (2015). A method to bias correct data with too little wet days is used in Cannon et al. (2015) and Vrac et al. (2016). Our approach for the handling of dry days is different but delivers very similar results.

Consequently, our paper provides an overview, an application, an evaluation and an upgrade of existing methods. We apply and compare eQC with other bias correction methods like QM and SDM. The latter one was used in the OEKS15 project in Austria.

The application of eQC (as a further development of EDCDFm and PresRAT) for Austria is new to the literature, and so is the combination of eQC with a method, that adds wet days if necessary. Another motivation for our work is the widely used dataset in Austria named OEKS15. We show errors in this dataset in the reference period compared to the observations that were used for bias correction.

**Specific comments**

Here are the answers to your specific concerns on after the other in the order they appeared in your review:

- *Moreover the text is written as if it was obvious that the CCS should be conserved by postprocessing methods, but there a reasonable arguments for and against this. Again, there is no justification at all for this in the manuscript and related literature is not sufficiently discussed.*
  Your concerns that the need to conserve the CCS is not obvious are now discussed in the paper in the introduction. We cite literature that discusses this topic and explain when it makes sense to conserve the CCS.

- 1) *The introduction cites a good number of relevant publications, but it is not well structured and many details are unclear. The attempt at grouping the methods in lines 67-81 is not clearly linked to the discussion of individual methods at the beginning of the introduction.*
  *The introduction should be organised from the beginning by explaining and giving*

*examples for groups of methods with common structural elements and properties, including*

*- Additive and multiplicative scaling, and linear regression*
*- Empirical QM, parametric QM using one function, multi-segment parametric QM*
*- Modification or preservation of the simulated CCS, specifying what aspect of the CCS is preserved, e.g. mean or CCS for specific quantiles*
*- Treatment of biases in wet-day frequency*
*- Treatment of values outside observed range*
*- Which methods assume stationarity of the bias? For those who don't, how do they specify potential non-stationarities?*

*It should also include a paragraph on comparison studies. This overview should include Maraun and Widmann (2018), which is a standard reference for downscaling and bias correction, and the VALUE comparison of downscaling and bias correction methods (Maraun et al. 2018, Gutierrez et al. 2019, Widmann et al. 2019).*
The introduction is now completely restructured following your suggestions. We replaced the part with the grouping of bias correction methods, as there are already more comprehensive examples in the literature (like Maraun and Widmann, 2018), with a table, where we group methods according to parametric or non-parametric and if the correction is applied on fixed values or fixed quantiles. The treatment of values outside the observed range is not a focus of our manuscript. So, in the sake of conciseness we skipped this discussion. We also added a paragraph that mentions overview and comparison studies including the four references in your list.

- 2) *The list of properties that the new method should satisfy (lines 97-100) is not clear. Property 1: What are 'long-term trends'? Differences between the simulated future and reference periods? If so, differences in the mean or in individual quantiles? Property 2: 'the model data' should be 'the corrected model data'. 'should match the observational data' with respect to what? Means, quantiles, trends?*
  The needed properties for our bias correction are now explained in more detail. Property 1: Long-term trends refers to the difference of reference period and simulated future period which is usually in the order of several decades. Property 2: To be exact we expect the corrected model data in the reference period to have the same climatological mean as the observations.

- 3) *The outline of the method (lines 107-112) is unclear. Up to here the text has mentioned as key issues the correction of an underestimation of wet days, and the preservation of the CCS. Now the question of whether biases are stationary is the key point, but this and the various ways of how to define QM have not been properly introduced. Most QM methods assume stationarity, not because it is necessarily a correct assumption, but because deviations from this have to have specific forms, for instance the one given in this manuscript, which are difficult to justify. As mentioned at the beginning of the review any specific approach needs of course to be justified.*
  Our presented bias correction is to a large extend similar to EDCDFm (Li et al., 2010) which was further developed to PresRAT by Pierce et al. (2015). Justification for

those methods are also found in those papers. Furthermore, we provide an example in section 3.1. where we show that EDCDFm sometimes leads to more plausible results than QM:

Consider the daily maximum temperatures for a grid point during a summer month in Europe, where the observations in the reference period are 20, 25, and 30 °C. The model simulates 20, 30 and 32 °C for the same period, i.e. there is a warm bias especially in the middle quantile. QM would suggest 0, -5 and -2 °C as bias for the model values. For the future period the model simulates 25, 35 and 36 °C. QM would correct this to 22.5, 33 and 34 °C. In this example, values in between are linearly interpolated, values above the range in the reference period (above 32 °C) are found through constant extrapolation, that is the correction value for the highest temperature also applies for even higher temperatures. eQC corrects at quantiles (not fixed values) which yields to 25, 30 and 34 °C. In this simple example, eQC seems to plausibly correct the model's warm bias at middle quantiles, while QM does not.

- 4) *Lines 102/103: the underestimation of the precipitation sums needs to be explained in more detail. If the full PDFs were matched the sums and thus the means would be identical by construction because mean(x) = ∫ x pdf(x) dx . If I understand correctly the problem is that the PDFs are not matched for the full range of values including zero, but only for the simulated wet days, which leads to the mean over the corrected simulated wet days matching the observed sum over these days, but the simulated mean over the whole period includes more dry days than the observed mean over the whole period.*

  The underestimation of precipitation sum is now explained in more detail in the introduction. Usually, if the bias is corrected multiplicatively, the zero-precipitation days have to be excluded from bias correction to avoid division by zero. This means that the mean over the corrected simulated wet days matches the observed sum over the exact same days, but there are more wet days in the observations.

- 5) *Whether the CCS of the original simulations should be preserved by bias correction is an important questions, but the correct answer depends on whether the raw CCS can be expected to be realistic. Discussions of this can be found for instance in Maraun and Widmann (2018), section 12.9.1, and in Maraun et al. (2017). The paper contains no justification for preserving the CCS.*

  A discussion section about whether the CCS should be conserved or not is now included in the introduction. In summary the modification of the CCS through bias correction can be a wanted feature, especially if the circulation patterns of the climate model are not plausible. Otherwise, it is sensible to want the CCS of the raw model to be unchanged.

- 6) *The validation of the OEKS15 data in section 4 is not convincing with the information available in the manuscript. In addition to structural elements in SDM that can lead to a difference in the means of the postprocessed models and the observations there are two more potential reasons for this difference. The first is the use of different observation datasets used for fitting SDM and for validation SDM. The*

*second is a different reference period for fitting SDM and for the validation. There is no information in the manuscript on these two points, and therefore it cannot be concluded that the differences that are found are due to the structure of SDM.*
We now provide further details on the comparisons of the OEKS15 data with the observations. The observations are GPARD data from ZAMG and the reference period is 1971-2000, as mentioned in Chimani et al. (2016). The errors found have to be attributed to the bias correction method itself.

Thank you again for all of these suggestions as they will clearly improve our manuscript and sharpen the novelty aspect.